# Surface Electromyography-Driven Parameters for Representing Muscle Mass and Strength

**DOI:** 10.3390/s23125490

**Published:** 2023-06-10

**Authors:** Joo Hye Sung, Seol-Hee Baek, Jin-Woo Park, Jeong Hwa Rho, Byung-Jo Kim

**Affiliations:** 1Department of Neurology, Korea University Anam Hospital, Korea University College of Medicine, Seoul 02841, Republic of Korea; centertruth@korea.ac.kr (J.H.S.); virgo3318@gmail.com (S.-H.B.); parkzinu@korea.ac.kr (J.-W.P.); iliebeme@naver.com (J.H.R.); 2BK21FOUR R&E Center for Learning Health Systems, Korea University, Seoul 02841, Republic of Korea

**Keywords:** electromyography, skeletal muscle mass, muscle strength, isometric exercise

## Abstract

The need for developing a simple and effective assessment tool for muscle mass has been increasing in a rapidly aging society. This study aimed to evaluate the feasibility of the surface electromyography (sEMG) parameters for estimating muscle mass. Overall, 212 healthy volunteers participated in this study. Maximal voluntary contraction (MVC) strength and root mean square (RMS) values of motor unit potentials from surface electrodes on each muscle (biceps brachii, triceps brachii, biceps femoris, rectus femoris) during isometric exercises of elbow flexion (EF), elbow extension (EE), knee flexion (KF), knee extension (KE) were acquired. New variables (MeanRMS, MaxRMS, and RatioRMS) were calculated from RMS values according to each exercise. Bioimpedance analysis (BIA) was performed to determine the segmental lean mass (SLM), segmental fat mass (SFM), and appendicular skeletal muscle mass (ASM). Muscle thicknesses were measured using ultrasonography (US). sEMG parameters showed positive correlations with MVC strength, SLM, ASM, and muscle thickness measured by US, but showed negative correlations with SFM. An equation was developed for ASM: ASM = −26.04 + 20.345 × Height + 0.178 × weight − 2.065 × (1, if female; 0, if male) + 0.327 × RatioRMS(KF) + 0.965 × MeanRMS(EE) (SEE = 1.167, adjusted R^2^ = 0.934). sEMG parameters in controlled conditions may represent overall muscle strength and muscle mass in healthy individuals.

## 1. Introduction

Skeletal muscle mass and strength are among the important indicators to represent health status in the elderly. Aging-related progressive skeletal muscle decrement starts in middle age and is associated with increased adverse health outcomes [1]. Especially, the value of appendicular skeletal muscle mass divided by the height square (ASM/height^2^) below the normal limit is used as a criterion for diagnosing sarcopenia [2]. The need for developing a simple and effective assessment tool for muscle mass has been increasing in a rapidly aging society. Nevertheless, the current standard methods for estimating muscle mass, such as dual-energy X-ray absorptiometry (DEXA), MRI, and bioimpedance analysis (BIA) with reliable accuracy, are expensive and have limited availability [3]. 

Surface EMG (sEMG) is a practical and cost-effective tool for measuring muscle activity and can be used during dynamic exercise. sEMG has been widely used to assess the electrophysiological processes of muscle contraction and force generation in research on sports and rehabilitation medicine [4]. In recent years, the applicability of sEMG as a method to provide early diagnosis and monitoring of sarcopenia has been increasingly investigated [5,6,7]. As sEMG records myoelectrical signals simply by attaching electrodes to the skin surface, the portable sEMG systems embedded into sportswear have been developed and have become popular in the sports and fitness industry. Smart sensors in combination with mobile apps and embedded systems which enable the monitoring of daily muscle activities will become more popular in the future healthcare industry [6,8]. With these technological innovations, it has become possible to easily collect more information about muscle activities through sEMG. However, the practical application of sEMG still has limitations due to the caution in the interpretation of sEMG data during exercise. 

Ultrasonography (US) has been frequently used in studies that require quantitative skeletal muscle assessment. US-derived muscle mass indicators were studied for sarcopenia diagnosis [9] and as a marker associated with functional outcomes such as the risk of falls in the geriatric population [10]. Another useful tool, BIA performs noninvasive body composition analysis, by measuring tissue conductivity after applying a weak electrical current. BIA has evolved over the past years to become a practical method to provide reliable parameters of low muscle mass, which has been incorporated into the recent diagnostic criteria of sarcopenia [2]. As variables obtained from BIA are also found to be associated with muscle function, BIA has become a promising tool for skeletal muscle evaluation [11]. 

Therefore, we aimed to investigate the feasibility of the sEMG parameters acquired in controlled laboratory conditions to estimate muscle mass and muscle strength. We used BIA to acquire the reference value of ASM. In addition, we analyzed the sEMG parameters in relation to various body composition data derived from BIA and muscle thickness measured using US.

## 2. Materials and Methods

### 2.1. Subjects and Study Design

We recruited healthy volunteers aged 40–80 years with normal muscle strength on manual muscle testing. Subjects with a neurological disorder or musculoskeletal disease that could have caused muscle weakness in the recent three months were excluded. In addition, subjects with muscle weakness as a sequela of previous neurological disorders or musculoskeletal disease before three months were excluded. The height and weight were measured in all participants, and the body mass index (BMI) was calculated. Each subject was asked to obtain BIA, muscle US, and sEMG data. Written informed consent was obtained from all participants. This study was approved by our institutional review board and conducted in accordance with the Declaration of Helsinki (No. 2020AN0361). 

Each study subject underwent all examinations with a one-day hospital visit. As exercise can cause fluid to shift to muscles [12], which may interfere with the results of BIA and muscle US, the study subjects were informed not to perform exercises before the examinations. For the same reason, muscle US and BIA were performed before the sEMG with exercise protocol was performed.

### 2.2. Bioimpedance Analysis

All the participants underwent BIA measurement (InBody 770; Inbody Corp, Seoul, Republic of Korea). By applying varying frequencies of alternating current through the body, BIA measures the tissue conductivity as the impedance and its two components: resistance and reactance. Using these variables in combination with other covariates, such as sex, weight, and height, BIA estimates the body composition according to the developed prediction equations. Thirty impedance measurements were taken at six different frequencies (1 kHz, 5 kHz, 50 kHz, 250 kHz, 500 kHz, and 1000 kHz) for five body segments (right and left arms, right and left legs, and trunk). The participants were instructed not to eat or exercise for at least three hours before the test and to maintain regular fluid intake the day before. BIA provides comprehensive information about body composition, which has been considered to have acceptable reliability and validity [13,14]. The measurement outputs such as ASM, body fat mass (BFM), segmental lean mass (SLM), and segmental fat mass (SFM) for the dominant side of the arm and leg were collected.

### 2.3. Ultrasound

The thickness of four muscles (biceps brachii, triceps brachii, rectus femoris, and biceps femoris) was measured on the dominant side using B-mode US imaging (Aplio i700, Canon, Otawara, Japan) with an 18-Mz linear array transducer (Figure 1). The participants were examined with full relaxation, shoulder and hip in the neutral position, and elbows and knees in full extension. The biceps brachii (BB), triceps brachii (TB), and rectus femoris (RF) were examined in the supine position, while the biceps femoris (BF) was examined in the prone position. Care was taken to apply minimal pressure on the skin and to keep the probe as perpendicular to the skin as possible throughout the examination. The muscle thickness was assessed using the following anatomical landmarks (BB, between the medial acromion and the cubital fossa at 1/3 distance from the cubital fossa; lateral head of TB, at the midpoint between the posterior crista of the acromion and the olecranon at 2 finger widths lateral to the line; RF, at the midpoint between the anterior superior iliac spine and superior aspect of the patella; and the long head of BF, at the midpoint between the ischial tuberosity and the lateral epicondyle of the tibia). The muscle thickness was measured at the maximal muscle bulk with the probe positioned on the transverse plane of the muscles. Muscle thickness refers to the distance between the deep and superficial aponeuroses of a muscle.

### 2.4. Exercise Protocol

The maximal voluntary contraction (MVC) strength was measured using a hand-held dynamometer (01165; Lafayette, IN, USA) during unilateral isometric exercises of elbow flexion (EF), elbow extension (EE), knee flexion (KF), and knee extension (KE) on the dominant side. During EF and EE, the subjects were placed in a supine position with an anatomical position on an examination bed. The elbow joint was flexed at a 90-degree angle, while the upper arm remained in contact with the bed. The hand-held dynamometer was placed on the forearm immediately proximal to the wrist joint. During the KF and KE, the subjects were in the prone position. The knee joint was flexed at a 90-degree angle, while the anterior thigh remained in contact with the bed. The hand-held dynamometer was placed on the shank, immediately proximal to the ankle joint. Just before exertion, the subjects were provided with instructions that “at the count of three, push/pull as hard and as fast as you can and hold that contraction”. The duration of each trial of the maximal voluntary exercise was 3 s. The tests were repeated to acquire three valid trials for each of the four exercises. The examiner paid special attention to minimizing changes in the position of the subject throughout the exam, and to succeed in overcoming the force produced by the subject during every trial for accurate MVC strength measurement. The maximum value among the three trials was used as MVC strength (i.e., MVCstrength(EF), MVCstrength(EE), MVCstrength(KF), and MVCstrength(KE)).

### 2.5. sEMG Activity Recording and Analysis

During MVC exercises, the sEMG signals were recorded using a standard EMG system (Synergy; Nicolet Biomedical, Madison, WI, USA). The surface electrodes (20 mm diameter; Natus Neurology, Middleton, WI, USA) were placed with the belly tendon method at five selected locations (#1, #2, #3, #4, and #5) in the dominant-side upper arm and six selected locations (#1, #2, #3, #4, #5, and #6) in the dominant-side thigh (Figure 2). The BB electrodes were placed on the line between the medial acromion and the cubital fossa at 1/3 distance from the cubital fossa. Three BB electrodes (#1, #2, #3) were positioned linearly and centered on the BB muscle belly without interelectrode distance. The TB electrodes were placed at the middle point on the line between the posterior crista of the acromion and the olecranon at 2 finger widths lateral to the line. Two TB electrodes (#4, #5) were positioned linearly and centered on the TB muscle belly without interelectrode distance. A neutral reference electrode was placed at the lateral epicondyle of the humerus. The RF electrodes were placed at the middle point on the line from the anterior superior iliac spine to the superior aspect of the patella. Three RF electrodes (#1, #2, #3) were positioned linearly and centered on the RF muscle belly with a 10 mm-interelectrode distance. The BF electrodes were placed at the middle point on the line between the ischial tuberosity and the lateral epicondyle of the tibia. Three BF electrodes (#4, #5, #6) were positioned linearly and centered on the BF muscle belly with a 10 mm interelectrode distance. A neutral reference electrode was placed at the lateral prominence of the patella. These methods were in accordance with SENIAM guidelines [15]. 

The sEMG signals were processed with a sampling frequency of 48 kHz. They were amplified using a differential amplifier (Natus, Nicolet Biomedical, Middleton, WI, USA) with a common mode rejection ratio of 110 dB and band-pass filtered from 100 to 500 Hz. The raw sEMG signals were full-wave-rectified. The mean and maximum values of the root mean square (RMS) from three electrodes (#1, #2, #3) during EF (MeanRMS(EF), MaxRMS(EF)), from two electrodes (#4 and #5) during EE (MeanRMS(EE), MaxRMS(EE)), from three electrodes (#1, #2, #3) during KE (MeanRMS(KE), MaxRMS(KE)), and from three electrodes (#4, #5, #6) during KF (MeanRMS(KF), MaxRMS(KF)) were acquired. Furthermore, the ratio of MeanRMS(EF) divided by the mean value of RMS from two electrodes (#4, #5) during EF, the ratio of MeanRMS(EE) divided by the mean value of RMS from three electrodes (#1, #2, #3) during EE, the ratio of MeanRMS(KE) divided by the mean value of RMS from three electrodes (#4, #5, #6) during KE, and the ratio of MeanRMS(KF) divided by the mean value of RMS from three electrodes (#1, #2, #3) during KF were calculated (RatioRMS(EF), RatioRMS(EE), RatioRMS(KE), RatioRMS(KF)). 

### 2.6. Statistical Analysis 

Descriptive statistics were used to depict the subjects’ characteristics, and all the data were presented as means and standard deviations. Pearson’s correlation coefficients were calculated between sEMG parameters, MVCstrength, BIA, and US parameters. Comparisons were always made between parameters derived during the same exercise maneuver or from the involved body segment (dominant side of the arm or leg) (e.g., MVCstrength (EF)-MeanRMS(EF)-MaxRMS(EF)-RatioRMS(EF)-SLM(arm)-SFM(arm)-muscle thickness(biceps brachii)). A stepwise linear regression analysis was used to determine the estimation models for ASM based on age, sex, height, weight, and sEMG parameters. The collinearity of the ASM estimation model was controlled using the Durbin–Watson test and the variance inflation factor. The subjects were randomly divided into 137 subjects in the model development group and 75 subjects in the cross-validation group according to the methodology of a previous study, where 30% of the total number of subjects were randomly extracted as a cross-validation group [16]. The estimation model was cross-validated in the cross-validation group. ASM from BIA and predicted ASM values were compared using Pearson’s correlation and paired *t*-tests. The analyses were performed using SPSS software (version 26.0, IBM, Armonk, NY, USA), and *p*-values < 0.05 were considered statistically significant.

## 3. Results

### 3.1. Subject Characteristics

We recruited a total of 212 participants (91 men and 121 women; age range, 41–79 years). The demographic and measurement data of the participants in the model development and cross-validation groups are presented in Table 1. Among them, 20 participants belonged to low muscle mass (ASM/height^2^ < 7.0 kg/m^2^ in men and <5.7 kg/m^2^ in women), and 11 participants were diagnosed with sarcopenia according to the updated sarcopenia definition by the Asian Working Group for Sarcopenia [2]. 

### 3.2. Correlation of sEMG with Other Parameters

The correlation analyses between sEMG parameters and MVCstrength are presented in Table 2. The sEMG parameters (MeanRMS, MaxRMS, RatioRMS) showed significant positive correlations with MVCstrength of each corresponding exercise. MVCstrength(EF), MVCstrength(EE), MVCstrength(KF), and MVCstrength(KE) showed the highest correlation coefficients with MaxRMS(EF) (r = 0.679, *p*-value < 0.01), MeanRMS(EE) (r = 0.531, *p*-value < 0.01), MeanRMS(KF) (r = 0.529, *p*-value < 0.01), and MeanRMS(KE) (r = 0.506, *p*-value < 0.01), respectively. 

Significant correlations were also observed among MVC strength, BIA, and US parameters (Table 2). MVCstrength showed positive correlations with SLM of related body segments and muscle thickness by US but showed negative correlations with SFM. MVCstrength(EF) showed positive correlations with the SLM of the dominant arm (r = 0.747, *p*-value < 0.01) and BB muscle thickness (r = 0.719, *p*-value < 0.01) and a negative correlation with SFM of the dominant arm (r = −0.279, *p*-value < 0.01). MVCstrength(EE) showed positive correlations with SLM of the dominant arm (r = 0.778, *p*-value < 0.01) and TB muscle thickness (r = 0.475, *p*-value < 0.01) and a negative correlation with SFM of the dominant arm (r = −0.258, *p*-value < 0.01). MVCstrength(KF) showed positive correlations with SLM of the dominant leg (r = 0.618, *p*-value < 0.01) and BF muscle thickness (r = 0.321, *p*-value < 0.01) and a negative correlation with SFM of the dominant leg (r = −0.177, *p*-value = 0.038). MVCstrength(KE) showed positive correlations with SLM of the dominant leg (r = 0.595, *p*-value < 0.01) and RF muscle thickness (r = 0.429, *p*-value < 0.01) and a negative correlation with SFM of the dominant leg (r = −0.195, *p*-value = 0.022). 

In addition, sEMG parameters were correlated with muscle mass and fat mass indicators (Table 3). All sEMG parameters were positively correlated with SLM, except MeanRMS(KE) and MaxRMS(EE, KE). The dominant arm SLM had the highest correlation coefficient with MaxRMS(EF) (r = 0.447, *p*-value < 0.01). The SLM of the dominant leg had the highest correlation coefficient with the RatioRMS(KF) (r = 0.366, *p*-value < 0.01). On the other hand, all sEMG parameters were negatively correlated with SFM, except MeanRMS(KE), MaxRMS(EE, KE), and RatioRMS(EF, KE). The dominant arm SFM had the highest correlation coefficient with MeanRMS(EE) (r = −0.392, *p*-value < 0.01). The dominant leg SFM had the highest correlation coefficient with MaxRMS(KF) (r = −0.416, *p*-value < 0.01).

All the sEMG parameters showed significant correlations with muscle thickness measured by US, except MeanRMS(KF), MaxRMS(KF) with BF thickness, and RatioRMS(EE) with TB thickness. The BB muscle thickness showed the highest positive correlation with MaxRMS(EF) (r = 0.472, *p*-value < 0.01). The TB muscle thickness showed the highest positive correlation with the MeanRMS(EE) (r = 0.260, *p*-value < 0.01). The BF muscle thickness was positively correlated with RatioRMS(KF) (r = 0.264, *p*-value < 0.01). The RF muscle thickness showed the strongest positive correlation with MaxRMS(KE) (r = 0.375, *p*-value < 0.01). 

Finally, correlation analyses were performed between the sEMG parameters and ASM, which showed significant correlations, except for MeanRMS(KE) and MaxRMS(KE). The correlation coefficients between ASM and sEMG parameters ranged from 0.170 (between ASM and MaxRMS(EE), *p*-value = 0.047) to 0.441 (between ASM and MaxRMS(EF), *p*-value < 0.01).

### 3.3. ASM Predicting Model

A stepwise multiple regression analysis was performed to produce an equation predicting the BIA-derived ASM based on the age, sex, height, weight, and sEMG parameters (MeanRMS, MaxRMS, and RatioRMS) (Table 4). Sex, height, weight, RatioRMS(KF), and MeanRMS(EE) remained as significant variables. An equation for estimating the ASM was developed as follows: *ASM* = −26.04 + 20.345 × *Height* + 0.178 × *weight* − 2.065 × (1, *if female*; 0, *if male*) + 0.327 × *RatioRMS(KF)* + 0.965 × *MeanRMS(EE) (SEE* = 1.167, *R*^2^ = 0.937, *Adjusted R*^2^ = 0.934) 

Additional regression models only with lower limb parameters (MeanRMS(KF, KE), MaxRMS(KF, KE), RatioRMS(KF, KE)) and upper limb parameters (MeanRMS(EF, EE), MaxRMS(EF, EE), RatioRMS(EF, EE)), respectively, are presented in the Appendix A.

The ASM prediction equation was applied to the cross-validation group. A strong correlation (r = 0.967, *p*-value < 0.001) was observed between the measured ASM and its predicted value (Figure 3). The predicted ASM did not significantly differ from the measured ASM using BIA (18.95 ± 4.33 vs. 19.18 ± 4.59; *p*-value = 0.101).

## 4. Discussion

In the present study, we developed an sEMG-based muscle mass estimation equation in a sample of healthy adults aged between 40 and 80 years. Newly developed sEMG parameters were positively correlated with muscle strength and mass indicators derived from BIA and US. In the estimation equation, sex, height, and weight were included as significant variables, in addition to sEMG-driven parameters. Although sex, height, and weight explain a large portion of the predicted muscle mass, sex and height are fixed variables for an individual under monitoring, and the weight is affected by various body compositions, including water, fat, as well as muscle. Therefore, sEMG parameters are the only variables in the estimation equation reflected by muscle mass.

Significant correlations were observed between the muscle strength and all RMS-based sEMG parameters. sEMG signals are produced by the summation of simultaneously evoked motor unit potentials. sEMG amplitudes (e.g., RMS) have been used to estimate the muscle force, as motor unit recruitment and firing rate, which are the two primary constituents of force generation, are reflected in the sEMG signals [17]. In our study, the RMS parameters accounted for up to 46% of the variance of the muscle strength, although the accountability varied widely according to the examined muscles. This variability occurs because the relationship between the RMS and produced force is not straightforward. Confounding factors must be considered when interpreting sEMG signals [18,19]. These factors include the anatomical and functional heterogeneity of individual muscles, such as muscle fiber orientation, the amount of tissue between the muscle and electrode, and cross-talk from the adjacent muscles. We selected four muscles (BB, TB, BF, and RF) to acquire sEMG signals for estimating the total muscle mass, which are the muscles involved in major joint movements during commonly performed exercises and placed superficially in a limb. As each muscle has its own intrinsic factors that influence sEMG signals, we attempted to determine the best combinations of muscles to acquire the sEMG signals used to predict the total muscle mass.

Furthermore, muscle strength showed significant positive correlations with muscle mass indicators and negative correlations with body fat mass indicators. It is widely accepted that muscle strength during maximal voluntary contraction depends on muscle mass [20,21,22,23]. Moreover, investigators have demonstrated the relationship between muscle strength and muscle cross-sectional area [24]. US has been increasingly used as a tool to measure muscle mass-related indicators [9,16,25,26,27,28,29,30,31]. The muscle thickness measured by US is closely related to the cross-sectional area measured by both US and computed tomography [32]. On the other hand, a higher body fat percentage is inversely associated with muscle strength [33,34,35]. This can be explained by the accumulation of intramuscular fat, which adversely affects muscle quality.

sEMG parameters showed positive correlations with SLM and muscle thickness and negative correlations with SFM. However, the two sEMG values acquired during KE (MeanRMS and MaxRMS) did not show any significant correlations. The lack of correlation between these parameters might be attributed to the anatomical characteristics of the quadriceps femoris, which is composed of four parts of large muscles situated at greater depths from the recording electrodes. This suggests that the attached electrodes could not provide representative information regarding the activity of the knee extensors as a whole. Notably, US measurements did not show a significant relationship between the BF thickness and the sEMG parameters. These results might have been caused by the technical difficulty of measuring the BF thickness using US, as it is relatively difficult to delineate the boundaries of the BF muscle.

Several studies have attempted to associate US-measured muscle thickness with sEMG-based muscle activation signals on different muscles, including the lower limb [28,36], upper arm [37,38], abdominal [25,26], and back muscles [39], which yielded inconsistent results depending on the slightly different definitions of the variables and exercise protocols. Brown and McGill^19^ did not observe a clear association between the abdominal muscle sEMG measurements and muscle thickening. The authors suggested that the negative result might be related to the composite laminate-like structure of the abdominal wall, which resulted in the improper transmission of sEMG signals to the electrodes on the skin. Unlike our study, most of these previous studies used the normalization method instead of raw sEMG values to compare muscles from different individuals and focused more on the thickness changes rather than the thickness itself. Their findings indicate that specific muscles are more appropriate for collecting representative sEMG signals to estimate muscle mass. Meanwhile, the influence of intramuscular fat and subcutaneous fat between the muscle and electrode on sEMG signals has been investigated [40,41]. Lanza et al. [42] found that the sEMG amplitudes of the hip abductor muscles may be reduced by intramuscular adipose tissue, which is consistent with our findings. They suggested that intramuscular fat tissue could affect the ability to activate muscle by changing the muscle architecture.

Finally, the relationship between the sEMG parameters and ASM was established. The RMS values acquired during EF had the highest correlation coefficients with muscle mass indicators, including ASM. All sEMG parameters except MeanRMS(KE) and MaxRMS(KE) demonstrated significant correlations with ASM. The relationship between sEMG RMS values and BIA parameters has been reported in previous studies [11,43]. The basic principle underlying BIA is to measure body electrical properties to estimate the extent of muscle membrane surface area filled with T-tubules, which correlates with muscle quantity. These acquired raw variables of BIA such as phase angle or cell membrane capacitance are regarded as not only muscle mass index but also muscle quality index as muscle cell integrity and cell function are also reflected in these variables. Several previous studies found that phase angle correlated with maximal muscle strength [44,45,46]. The muscle cell membrane is the source of EMG activity signals. Therefore, EMG parameters are affected at least in part by cellular membrane function reflected in BIA parameters. This was proven in a previous study that showed membrane capacitance of the leg assessed by bioelectrical impedance spectroscopy was associated with contractile properties and sEMG RMS parameters of plantar flexors during maximal voluntary contraction [43].

Among the introduced sEMG variables, RatioRMS(KF) and MeanRMS(EE) were included in the prediction equation. This means that sEMG signals recorded from the BF and TB play a significant role in predicting the total muscle mass. Although sEMG parameters acquired from BB during EF had the highest correlation coefficient with ASM, after sex, weight, and height were taken into account, they were excluded from the equation. Sarcopenia is known to mainly affect the muscles of the lower limb rather than those of the upper limb, which is called “regional” or “site-specific” sarcopenia [47]. Therefore, including the sEMG variable derived from the lower limb seems reasonable. In a previous study [48], the thicknesses of various muscles of the body were measured and compared between young (<50 years) and old (≥50 years) age groups. The study found that the thicknesses of the thigh and triceps muscles were significantly different between the two groups, which is in line with our results.

We used BIA and US, which enable regional assessments of muscles and body composition, to investigate factors influencing the sEMG activity. In particular, BIA is superior in performing a body composition analysis according to each body compartment. Based on these results, we examined the applicability of sEMG for skeletal muscle evaluation. However, this study had some limitations. This estimation equation was developed for healthy individuals without abnormal local muscle atrophy. It may also be difficult to apply this method to people who are very young or old. In this study, we selected four specific muscles. If more muscles are evaluated, the prediction model could be further improved; however, it would be less practical. Additionally, we used a hand-held dynamometer for maximal strength assessment instead of a fixed laboratory-based dynamometry, which is the gold standard assessment tool for maximal voluntary isometric contraction strength. However, if used with standardized techniques to minimize potential limitations, a hand-held dynamometer can establish reliability and validity [49,50]. Finally, we used the linear regression model to develop an ASM prediction equation, which was the statistical method used in the previous studies [9,16,51,52,53] with similar study objects. However, caution should be applied in the interpretation of the final result. As various muscle parameters of a single subject are, strictly speaking, not independent, it is possible to overestimate the performance of the prediction equation.

## 5. Conclusions

In conclusion, the sEMG parameters could represent overall muscle strength and muscle mass in healthy individuals. The RMS values acquired from the BF during KF and from the TB during EE can be used as possible parameters representing an individual’s muscle mass. The final regression model using sex, height, weight, and two sEMG parameters explained 93.4% of the variance of ASM. Although the application of sEMG in the real world beyond research needs further development regarding advanced signal processing techniques and validation processes due to inherent limitations of sEMG technology, it may be a possible assessment tool for muscle mass in healthy middle-aged and old-aged adults.

## Figures and Tables

**Figure 1 sensors-23-05490-f001:**
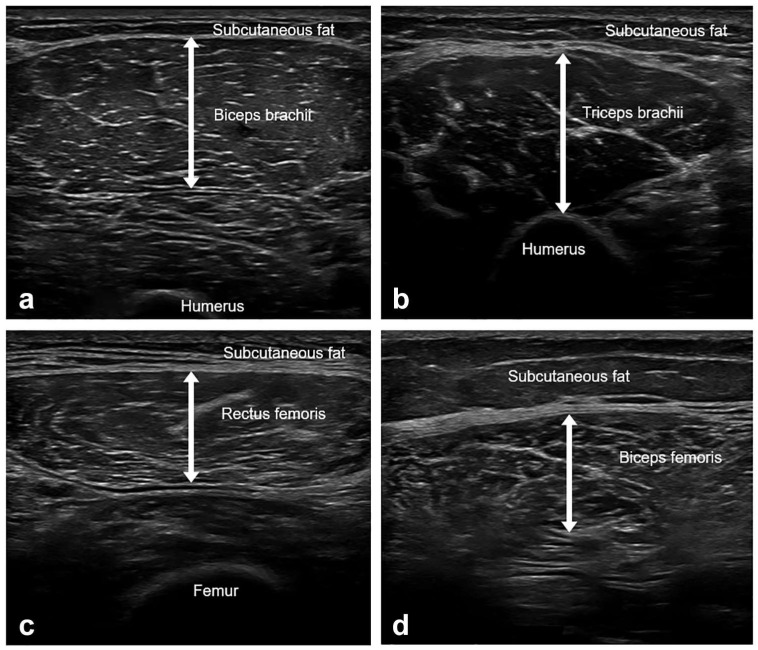
Representative images of muscle ultrasound and muscle thickness measurements. (**a**) Biceps brachii, between the medial acromion and the cubital fossa at 1/3 distance from the cubital fossa. (**b**) Triceps brachii, at the midpoint between the posterior crista of the acromion and the olecranon at 2 finger widths lateral to the line. (**c**) Rectus femoris, at the midpoint between the anterior superior iliac spine and the superior aspect of the patella. (**d**) Biceps femoris, at the midpoint between the ischial tuberosity and the lateral epicondyle of the tibia. Muscle thickness was defined as the distance between the deep and superficial aponeurosis of the muscle. The measurements were taken at their maximal muscle bulk.

**Figure 2 sensors-23-05490-f002:**
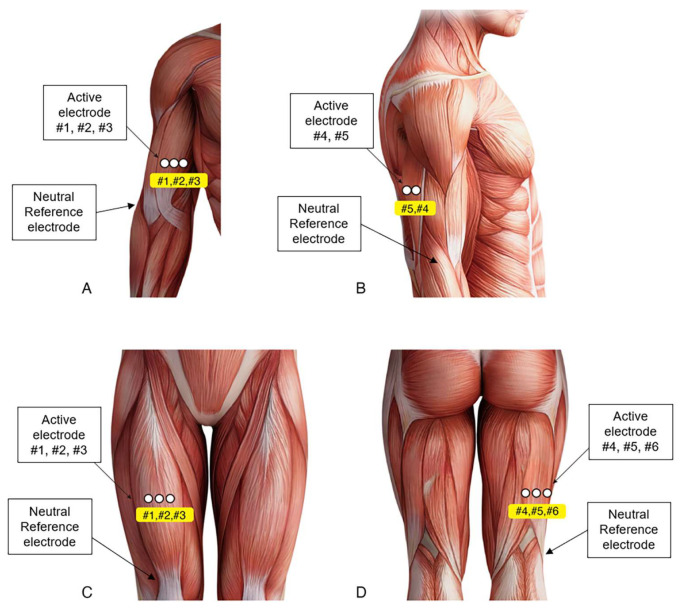
Placement of surface electromyography (sEMG) electrodes. The active electrodes were placed at five selected locations (#1, #2, #3, #4, #5) in the dominant-side upper arm (**A**,**B**), and six selected locations (#1, #2, #3, #4, #5, #6) in the dominant-side thigh (**C**,**D**). Among the five electrodes in the upper arm, three active electrodes (#1, #2, #3) were positioned linearly and centered on the biceps brachii (BB) muscle belly without interelectrode distance (**A**). Two electrodes (#4, #5) were positioned linearly and centered on the triceps brachii (TB) muscle belly without interelectrode distance (**B**). A neutral reference electrode was placed at the lateral epicondyle of the humerus. Among the six electrodes in the thigh, three electrodes (#1, #2, #3) were positioned linearly and centered on the rectus femoris (RF) muscle belly with a 10 mm interelectrode distance (**C**). The other three electrodes (#4, #5, #6) were positioned linearly and centered on the biceps femoris (BF) muscle belly with a 10 mm interelectrode distance (**D**). A neutral reference electrode was placed at the lateral prominence of the patella.

**Figure 3 sensors-23-05490-f003:**
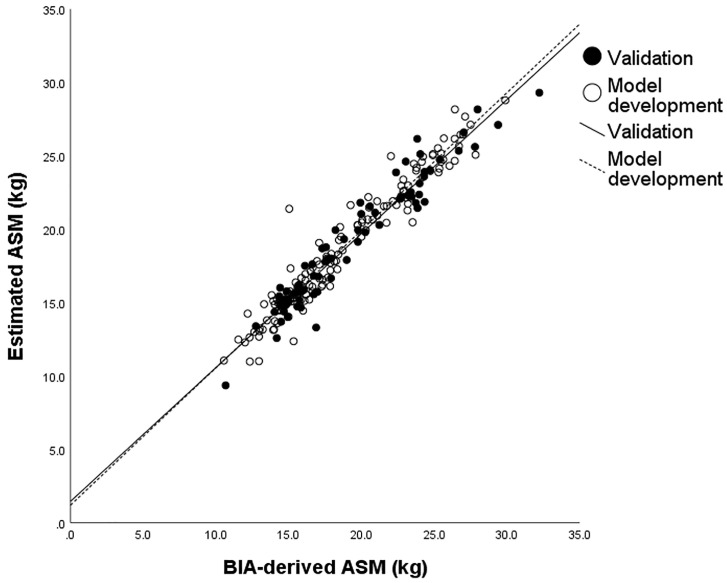
Correlation of ASM predicted by sEMG parameters with ASM measured by BIA. The white circles belong to the model development group and the black circles belong to the cross-validation group (r = 0.967, *p* < 0.001). ASM, appendicular skeletal muscle mass; BIA, bioimpedance analysis.

**Table 1 sensors-23-05490-t001:** Demographic and physical characteristics of the study subjects.

	Total (*n* = 212)	Model Development (*n* = 137)	Cross-Validation (*n* = 75)
Age	59 (11)	61 (10)	57 (10)
Female (*n*, %)	121 (56.5)	78 (56.9)	43 (57.3)
Height (m)	1.62 (0.08)	1.62 (0.09)	1.63 (0.08)
Weight (kg)	64.6 (11.2)	64.5 (10.8)	64.9 (12.0)
BMI (kg/m^2^)	24.4 (3.0)	24.4 (3.0)	24.3 (3.1)
ASM (kg)	18.95 (4.56)	18.83 (4.56)	19.18 (4.59)
BFM (kg)	18.75 (5.84)	18.85 (6.23)	18.55 (5.10)
MVCstrength(EF) (kg)	19.7 (6.2)	19.6 (6.3)	19.9 (6.2)
MVCstrength(EE) (kg)	13.8 (4.3)	13.9 (4.5)	13.6 (4.0)
MVCstrength(KF) (kg)	13.0 (4.3)	12.8 (4.2)	13.3 (4.3)
MVCstrength(KE) (kg)	24.0 (8.2)	24.2 (8.4)	23.6 (7.9)
MeanRMS(EF) (mV)	0.66 (0.27)	0.66 (0.28)	0.65 (0.24)
MeanRMS(EE) (mV)	0.73 (0.30)	0.74 (0.30)	0.70 (0.30)
MeanRMS(KF) (mV)	0.32 (0.13)	0.32 (0.13)	0.32 (0.13)
MeanRMS(KE) (mV)	0.23 (0.10)	0.23 (0.10)	0.23 (0.10)
MaxRMS(EF) (mV)	0.72 (0.30)	0.72 (0.31)	0.73 (0.28)
MaxRMS(EE) (mV)	0.31 (0.13)	0.30 (0.13)	0.33 (0.14)
MaxRMS(KF) (mV)	0.37 (0.16)	0.37 (0.16)	0.38 (0.16)
MaxRMS(KE) (mV)	0.25 (0.11)	0.25 (0.10)	0.26 (0.11)
RatioRMS(EF)	2.30 (0.69)	2.36 (0.73)	2.20 (0.61)
RatioRMS(EE)	2.44 (0.42)	1.44 (0.38)	1.43 (0.48)
RatioRMS(KF)	2.53 (0.98)	2.57 (0.96)	2.45 (1.01)
RatioRMS(KE)	1.88 (0.63)	1.87 (0.57)	1.90 (0.71)
BB thickness (mm)	13.91 (3.03)	13.82 (2.84)	14.08 (3.36)
TB thickness (mm)	10.61 (3.51)	10.52 (3.52)	10.79 (3.51)
BF thickness (mm)	19.09 (4.51)	19.05 (4.35)	19.15 (4.81)
RF thickness (mm)	11.26 (2.51)	11.11 (2.45)	11.53 (2.61)

Data are presented as mean (SD). MVCstrength: maximal voluntary contraction strength during isometric exercise. BMI, body mass index; ASM, appendicular skeletal muscle mass; BFM, body fat mass; RMS, root mean square; EF, elbow flexion; EE, elbow extension; KF, knee flexion; KE, knee extension; BB, biceps brachii; TB, triceps brachii; BF, biceps femoris; RF, rectus femoris.

**Table 2 sensors-23-05490-t002:** Correlation coefficients between MVC strength and RMS of sEMG parameters, US, and BIA parameters.

	MeanRMS	MaxRMS	RatioRMS	SLM (kg)	SFM (kg)	Muscle Thickness (mm)
MVCstrength(EF) (kg)	0.671 **	0.679 **	0.412 **	0.747 **	−0.279 **	0.719 **
MVCstrength(EE) (kg)	0.531 **	0.288 **	0.400 **	0.778 **	−0.258 **	0.475 **
MVCstrength(KF) (kg)	0.529 **	0.505 **	0.266 **	0.618 **	−0.177 *	0.321 **
MVCstrength(KE) (kg)	0.506 **	0.480 **	0.390 **	0.595 **	−0.195 *	0.429 **

MVCstrength: maximum voluntary contraction strength during isometric exercise. Comparisons were always made between parameters derived during the same exercise maneuver or from the involved body segment (dominant side of the arm or leg) (e.g., MVCstrength(EF)-MeanRMS(EF)-MaxRMS(EF)-RatioRMS(EF)-SLM(arm)-SFM(arm)-muscle thickness(biceps brachii)). ** *p*-value< 0.01 * *p*-value< 0.05. MVC, maximum voluntary contraction; RMS, root mean square; US, ultrasonography; BIA, bioimpedance analysis; EF, elbow flexion; EE, elbow extension; KF, knee flexion; KE, knee extension.

**Table 3 sensors-23-05490-t003:** Correlation coefficients between RMS of sEMG parameters and US and BIA parameters.

	SLM (kg)	SFM (kg)	Muscle Thickness (mm)	ASM (kg)
MeanRMS(EF) (mV)	0.432 **	−0.347 **	0.457 **	0.428 **
MeanRMS(EE) (mV)	0.351 **	−0.392 **	0.260 **	0.377 **
MeanRMS(KF) (mV)	0.270 **	−0.393 **	0.151	0.235 **
MeanRMS(KE) (mV)	0.154	−0.162	0.363 **	0.136
MaxRMS(EF) (mV)	0.447 **	−0.357 **	0.472 **	0.441 **
MaxRMS(EE) (mV)	0.158	−0.211	0.213 *	0.170 *
MaxRMS(KF) (mV)	0.229 **	−0.416 **	0.146	0.203 *
MaxRMS(KE) (mV)	0.129	−0.166	0.375 **	0.114
RatioRMS(EF) (mV)	0.341 **	−0.140	0.297 **	0.329 **
RatioRMS(EE) (mV)	0.274 **	−0.249 **	0.164	0.297 **
RatioRMS(KF) (mV)	0.366 **	−0.339 **	0.264 **	0.388 **
RatioRMS(KE) (mV)	0.220 **	−0.133	0.324 **	0.218 *

Comparisons were made between the parameters derived during the same exercise maneuver or from the involved body segment (dominant side of the arm or leg) (e.g., MVCstrength (EF)-MeanRMS(EF)-MaxRMS(EF)-RatioRMS(EF)-SLM(arm)-SFM(arm)-muscle thickness(biceps brachii)). ** *p*-value< 0.01 * *p*-value< 0.05. RMS, root mean square; US, ultrasonography; BIA, bioimpedance analysis; SLM, segmental lean mass; SFM, segmental fat mass; ASM, appendicular skeletal muscle mass; EF, elbow flexion; EE, elbow extension; KF, knee flexion; KE, knee extension.

**Table 4 sensors-23-05490-t004:** Stepwise regression analysis for ASM.

Entered predictor variables	Equation: *ASM* = −26.04 + 20.345 × *Height* + 0.178 × *weight* − 2.065 × (1, *if, female*; 0, *if male*) + 0.327 × *RatioRMS(KF)* + 0.965 × *MeanRMS(EE)*				
Age, sex, height, weight, MeanRMS(EF, EE, KF, KE), MaxRMS(EF, EE, KF, KE), RatioRMS(EF, EE, KF, KE)		*β*	Standard error	VIF	*p*-value
	Constant	−26.040	3.155		
	Height (m)	20.345	2.058	3.162	<0.001
	Weight (kg)	0.178	0.013	1.984	<0.001
	Sex (female)	−2.065	0.338	2.824	<0.001
	RatioRMS(KF) (mV)	0.327	0.118	1.275	0.006
	MeanRMS(EE) (mV)	0.965	0.373	1.270	0.011
	R^2^	0.937			
	Adjusted R^2^	0.934			
	SEE	1.167			
	Durbin–Watson statistic	1.844			

ASM, appendicular skeletal muscle mass; RMS, root mean square; EF, elbow flexion; EE, elbow extension; KF, knee flexion; KE, knee extension; SEE, standard error of estimate; VIF, variance inflation factor.

## Data Availability

The datasets analyzed during the current study are available from the corresponding author upon reasonable request.

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
