# Peer review of "Surface Electromyography-Driven Parameters for Representing Muscle Mass and Strength"

_sensors, 2023, doi:10.3390/s23125490_

Round 1
Reviewer 1 Report
Dear Authors,
This study aimed to evaluate feasibility of the surface electromyography (sEMG) parameters for estimating muscle mass. This is important in assessing sarcopenia associated with both aging and disease. The manuscript is well written, but some revisions are needed to improve its readability. The main problem of the methodology of this study is the processing of EMG.
1. In the "Methods" section, from the paragraph 2.4 it is not clear how the MVS was measured using hand-held dynamometer during elbow flexion (EF), elbow extension (EE), knee flexion (KF), and knee extension (KE).
2. The paragraph 2.5 is poorly readable due to the large number of abbreviations (names of muscles, EMG parameters, etc.). A figure explaining the placement of the electrodes is recommended.
3. The EMG processing was done roughly. Why were such filtering options chosen? In EMG processing, why cut-off frequency was established at 100 Hz? The important information on motor unit activity founds itself between 8 and 15 Hz. In EMG studies, 10 or even 5 Hz is generally recommended.
4. Statistical analysis and types of comparison should be presented more clearly.
5. In table 1 BMI (kg/Ht2) should be kg/m2.
6. Please check all abbreviations including the following:
Line 31 – explain the index ASM/ht2
Line 50 - appendicular skeletal muscle mass (ASM) - repeated abbreviation of words
Line 52, 61, 67, 85 etc.- ultrasonography (US)
My overall comment, the manuscript is not ready for publication in its present form, as it has flaws and needs to be edited.
The manuscript is well written, with some typos needed to be corrected.
Reviewer 2 Report
This is a well designed and well performed study on how to estimate total body muscle mass in an easy and reliable way. It certainly fits the scope of the journal. There is some room for improvement:
Obviously the various muscle parameters of a single participant are considered as independent in the analysis. This is certainly not the case, so the various linear models that were calculated by the authors should be replaced by a mixed effects model with the participant as random effect. Otherwise the quality of the model to estimate ASM is over optimistic.
The authors present many correlation coefficients on “global” parameters, e. g. between elbow flexion and appendicular muscle mass. This might be useful, but the parameters assessed are related to a single muscle, so the correlation coefficients should be presented, at least in addition, with regard to the single muscle, e.g. between MeanRMS biceps brachii and muscle thickness, because this is the unit of observation.
The authors do not present the various models tested in order to achieve the final model. This information should be given, at least in a supplement, github repository or something like that. Especially, the difference between the final model and a model that takes only the muscle parameters of the legs into account is of interest. Which criterion was used in the stepwise regression analysis?
good
Reviewer 3 Report
In this study the authors aimed to investigate the feasibility of the surface electro- myography (sEMG) parameters for estimating muscle mass.
Although the study has the potentiality of being shared with the scientific community, I believe that the manuscript would benefit from a minor revision with the attempt to better support their experimental setting.
1. the theoretical framework is scarce, they should clearly describe the scientific evidence that supports the hypothesis they have raised.
2. A lot of necessary information is missing in methods section:
- Experimental procedures should be better defined
- More information should be provided about the participants’ characteristics.
- The intervention protocol should be better described.
This element is missing from the methodological description, which may imply an impossibility of replicating the study due to a lack of clarity in this regard.
3. The Discussion should be enriched with the existing theory. The authors should clearly describe the scientific evidence that supports their findings. In addition, they should start with a first paragraph describing the main aims and then the main results.
Sincerely
Round 2
Reviewer 1 Report
The manuscript has been sufficiently improved to warrant publication in Sensors
Minor editing of English language required
Author Response
Thank you for your review.
Reviewer 2 Report
The present version has substantially improved. One statistical issue is still to be solved: In your answer to my first point you addressed the aspect of collinearity, which is well covered.
The point I made addresses stochastic independence. Since you assessed more than one muscle in one participant, the "added" muscle in the same participant adds less information than the same muscle in another participant, because they are not independent. But independence is a major statistical requirement for linear regression analysis. There are methods to correct for that (linear mixed effect models). Please correct that, otherwise you underestimate variance and errors.
